# Concurrent Cetuximab and Nivolumab as a Second-Line or beyond Treatment of Patients with Recurrent and/or Metastatic Head and Neck Squamous Cell Carcinoma: Results of Phase I/II Study

**DOI:** 10.3390/cancers13051180

**Published:** 2021-03-09

**Authors:** Christine H. Chung, Marcelo Bonomi, Conor E. Steuer, Jiannong Li, Priyanka Bhateja, Matthew Johnson, Jude Masannat, Feifei Song, Juan C. Hernandez-Prera, Bruce M. Wenig, Helen Molina, Joaquim M. Farinhas, Caitlin P. McMullen, J. Trad Wadsworth, Krupal B. Patel, Julie A. Kish, Jameel Muzaffar, Kedar Kirtane, James W. Rocco, Michael J. Schell, Nabil F. Saba

**Affiliations:** 1Department of Head and Neck-Endocrine Oncology, Moffitt Cancer Center, Tampa, FL 33612, USA; Matthew.Johnson@moffitt.org (M.J.); jude.masannat@moffitt.org (J.M.); Feifei.song@moffitt.org (F.S.); Caitlin.McMullen@moffitt.org (C.P.M.); Trad.Wadsworth@moffitt.org (J.T.W.); Krupal.Patel@moffitt.org (K.B.P.); Jameel.muzaffar@moffitt.org (J.M.); Kedar.kirtane@moffitt.org (K.K.); 2Department of Internal Medicine, The Ohio State University, Columbus, OH 43210, USA; marcelo.bonomi@osumc.edu (M.B.); Priyanka.Bhateja@osumc.edu (P.B.); 3Department of Hematology and Medical Oncology, Winship Cancer Institute, Emory University, Atlanta, GA 30322, USA; csteuer@emory.edu (C.E.S.); Nfsaba@emory.edu (N.F.S.); 4Department of Biostatistics and Bioinformatics, Moffitt Cancer Center, Tampa, FL 33612, USA; Jiannong.Li@moffitt.org (J.L.); michael.schell@moffitt.org (M.J.S.); 5Department of Pathology, Moffitt Cancer Center, Tampa, FL 33612, USA; Juan.Hernandez-Prera@moffitt.org (J.C.H.-P.); Bruce.wenig@moffitt.org (B.M.W.); Helen.Molina@moffitt.org (H.M.); 6Department of Radiology, Moffitt Cancer Center, Tampa, FL 33612, USA; Joaquim.Farinhas@moffitt.org; 7Department of Personalized Medicine, Moffitt Cancer Center, Tampa, FL 33612, USA; Julie.kish@moffitt.org; 8Department of Otolaryngology, The Ohio State University, Columbus, OH 43210, USA; James.rocco@osumc.edu

**Keywords:** cetuximab, nivolumab, pembrolizumab, real world data, HNSCC

## Abstract

**Simple Summary:**

To improve overall survival (OS), we evaluated a combination of cetuximab and nivolumab for toxicity and efficacy in patients with incurable recurrent and/or metastatic head and neck squamous cell carcinoma (HNSCC). In addition, electronic health record-derived real-world data were used to provide clinical context for our prospective findings and to explore sequential treatment effects of cetuximab and immune checkpoint inhibitors (CPI) including nivolumab and pembrolizumab. The cetuximab and nivolumab combination was very well tolerated. Although patients with no prior CPI showed a trend for more favorable progression-free survival relative to patients with prior CPI, the improvement in the 1-year OS did not reach the statistical threshold in this heavily treated patients. Optimal sequencing of cetuximab and CPI may have an impact in prognosis and requires further evaluation in management of patients with incurable HNSCC.

**Abstract:**

We hypothesized the combination of cetuximab and nivolumab would improve survival in recurrent and/or metastatic (R/M) HNSCC by providing synergy in cancer control and evaluated toxicities and efficacy of the combination. Effects of sequential administration of cetuximab and anti-Programmed Cell Death-1 checkpoint inhibitors (CPI) were also explored. Patients who failed at least one line of palliative treatment for incurable HNSCC were treated with cetuximab 500 mg/m^2^ IV on Day (D)-14 as a lead-in followed by cetuximab 500 mg/m^2^ IV and nivolumab 240 mg/m^2^ IV on D1 and D15 every 28-D cycle. Electronic health record-derived real-world data (RWD) were used to explore sequential treatment effects of CPI and cetuximab. A total of 45 evaluable patients were analyzed, and 31/45 (69%) patients had prior exposure to either CPI or cetuximab. The only grade 4 treatment-related adverse event was cetuximab infusion reaction in one patient. The 1-year progression-free survival (PFS) and overall survival (OS) rates were 19% and 44%, respectively. Although patients with no prior CPI (23/45, 51%) showed a trend for more favorable PFS relative to patients with prior CPI (22/45, 49%), the improvement in the 1-year OS did not reach the statistical threshold. For evaluation of sequential CPI and cetuximab treatment effects, we selected RWD-cetuximab cohort with 173 patients and RWD-CPI cohort with 658 patients from 6862 R/M HNSCC. Our result suggested patients treated with RWD-cetuximab after RWD-CPI had worse OS compared to no prior RWD-CPI (HR 1.81, 95% CI 1.02–3.16). Our data suggest the combination of cetuximab and nivolumab is well tolerated. Optimal sequencing of cetuximab and CPI may have an impact in prognosis and requires further evaluation.

## 1. Introduction

Head and neck squamous cell carcinoma (HNSCC) arise from the mucosal layers within the head and neck region. The most common risk factors are tobacco use and human papillomavirus (HPV) infection [1,2,3]. Patients with HPV-related HNSCC have more favorable prognosis compared to those with HPV-unrelated HNSCC [4]. While newly diagnosed, locally advanced HNSCC is highly curable with multi-modality therapies, recurrent and/or metastatic (R/M) HNSCC remains as an incurable disease [4,5,6,7,8,9]. Therefore, development of novel therapies with improved efficacy without significant toxicities is urgently needed for this patient population. 

The most recent advancement in the management of R/M HNSCC is in development of anti-Programmed Cell Death-1 (PD-1) checkpoint inhibitors (CPI), resulting in activation of T-cell dependent adaptive immunity [10,11]. Two randomized phase III clinical trials comparing nivolumab or pembrolizumab against investigator’s choice (IC) standard of care agents (methotrexate, docetaxel, or cetuximab) showed superior efficacy of nivolumab or pembrolizumab, with more favorable toxicity profiles and quality of life in platinum-refractory R/M HNSCC [7,8,12,13]. However, only limited numbers of patients gain clinical benefit. For example, treatment with nivolumab or pembrolizumab has a median overall survival (OS) of 7.5–8.4 months in platinum refractory R/M HNSCC [7,14]. In addition, a randomized phase III clinical trial comparing two experimental arms [pembrolizumab monotherapy or a combination of pembrolizumab, cisplatin, and 5-fluorouracil (5-FU)] against a combination of cetuximab, cisplatin, and 5-FU (EXTREME regimen) for first line therapy for incurable R/M HNSCC patients showed pembrolizumab-containing arms were superior to the EXTREME regimen [15]. In the total cohort of untreated incurable R/M HNSCC, the median OS of pembrolizumab monotherapy was 11.6 months, and the median OS of pembrolizumab with chemotherapy was 13.0 months compared to the median OS of 10.7 months in the EXTREME arm [15]. 

While there is a clear evidence that the combination of chemotherapy and CPI improves OS in R/M HNSCC, there are also significant toxicities associated with the chemotherapy [15]. It would be beneficial to evaluate less toxic and potentially synergistic targeted agents as a combination partner with CPI or sequence them to prime the tumor immune microenvironment (TIME) to enhance the efficacy of CPI. Epidermal growth factor receptor (EGFR) is expressed in the majority of HNSCC, and overexpression is associated with a poor prognosis [16,17,18]. Cetuximab, anti-EGFR antibody, is the only FDA-approved targeted agent used in HNSCC as a monotherapy, concurrent with radiation, or in combination with platinum-based chemotherapy [19]. Although EGFR is expressed in >90% of HNSCC, cetuximab has a modest response rate of 13% as a monotherapy in R/M HNSCC [16,20]. The known mechanisms of action are through inhibition of EGFR signal transduction in the cancer cells as well as activation of natural killer (NK) cell-mediated antibody dependent cell mediated cytotoxicity (ADCC) inducing innate immune response [21,22]. However, cetuximab is also known to have immunosuppressive effects by promoting expansion of regulatory T cells (Treg) in the TIME [23]. Given the immunomodulatory effects of cetuximab, evaluations of a combination of cetuximab and CPI and potential effects of sequential administration of CPI and cetuximab are warranted [24]. 

In this study, we hypothesized the combination of cetuximab and nivolumab would improve OS in R/M HNSCC by activating both innate and T-cell dependent adaptive immune response, providing synergy in cancer control [21,22]. In addition, two contemporary R/M HNSCC cohorts treated with either cetuximab monotherapy or nivolumab/pembrolizumab monotherapy selected from a de-identified electronic health record-derived real-world dataset (RWD) were analyzed to provide a clinical context for our prospective clinical trial results of the combination therapy. Furthermore, potential effects of sequential treatments with cetuximab monotherapy (RWD-cetuximab) and nivolumab/pembrolizumab CPI monotherapy (RWD-CPI) were also explored using the RWD cohort.

## 2. Results

A total of 47 patients were enrolled from three institutions during December 2017–May 2019, and 45 patients were deemed to be evaluable, as the other two did not receive any study treatment (Appendix A). One patient did not receive C1D1 of cetuximab and nivolumab because the treatment was delayed for >4 weeks due to an adverse event (Grade 3 fatigue) after the lead-in cetuximab. One patient expired at three days with unknown cause after receiving the lead-in cetuximab and did not receive C1D1 of cetuximab and nivolumab.

### 2.1. Phase I Safety Evaluation

The first three evaluable patients treated at Dose Level 1 completed the DLT evaluation period. No DLT was observed during the 4 weeks of observation period after C1D1. No dose reduction was required. As such, Dose Level 1 was determined to be the recommended phase II dose of lead-in cetuximab 500 mg/m^2^ alone (Day-14 before Cycle 1 only) followed by nivolumab 240 mg IV + cetuximab 500 mg/m^2^ every 2 weeks for 24 cycles or discontinuation. The patients treated at Dose Level 1 in the phase I portion of the study were included as a part of the phase II patient population.

### 2.2. Phase II Patient Characteristics

A total of 45 evaluable patients were analyzed, and patient characteristics were summarized in Table 1. The most common disease site was oropharynx or unknown primary (26/45, 58%); 22 were p16-positive (85%) and 4 were p16-negative (15%). PD-L1 CPS was evaluable in 39/45 (87%): 7 CPS < 1 (16%), 13 CPS 1–19 (29%), and 19 CPS ≥ 20 (42%). The smoking status was 6 current (13%), 27 former (60%), and 12 never (27%), with median pack years of 20 (range 0–185). 

Prior chemotherapy was given in 44 (98%). Radiotherapy was given with curative intent in 43/45 (96%) patients with platinum in 36/43 (84%), cetuximab in 3/43 (7%), and docetaxel in 1/43 (2%). Only one patient was platinum naïve because the patient presented with distant metastatic disease at the initial diagnosis and received ipilimumab and nivolumab as the first line recurrent/metastatic treatment. During the entire course of the treatments, 31/45 (69%) patients had prior exposure to either CPI or cetuximab. Detailed prior treatment descriptions were summarized in Appendix A.

### 2.3. Phase II Safety Evaluation

The complete TRAE and IRAE were summarized in Appendix A. The most common grade 3 TRAEs were fatigue (6/45, 13.3%) and skin toxicities (4/45, 8.8%) (Table 2). The only grade 4 TRAE was cetuximab infusion reaction in one patient (2.2%). The most common grade 3 IRAE was fatigue (3/45, 6.7%) while there was no grade 4 IRAE (Table 3). No grade 5 TRAEs or IRAEs occurred. Only grade 4 cetuximab infusion reaction resulted in discontinuation of the treatment.

### 2.4. Phase II Efficacy Evaluation

The overall response rate was 22.2%: 2 (4.4%) complete response, 9 (17.8%) partial response, 19 (42.2%) stable disease, and 16 (35.6%) progressive disease (Figure 1A,B). The median follow-up time for survival analyses of the study was 21.0 months calculated by the reverse KM method (data locked on 26 October 2020). The median progression-free survival (PFS) and 1-year PFS were 3.4 months and 19% (Figure 2A). The median OS and 1-year OS were 9.7 months and 44% (Figure 2B). Although some of the patients who were treated CPI prior to the study enrollment had objective responses, overall patients who did not receive CPI prior to receiving the cetuximab and nivolumab combination showed a trend to have better survival outcomes than patients who received prior CPI (Figure 2C,D). The PD-L1 CPS (<20 vs. ≥20) and p16 status did not associate with survival outcomes (Appendix A).

### 2.5. Contemporary Real-World Data Cohort Analyses

To provide a clinical context for the degree of the combination efficacy, we obtained a nationwide electronic health record-derived RWD from 6862 de-identified R/M HNSCC patients and selected two contemporary cohorts of patients who were treated with either cetuximab monotherapy or nivolumab/pembrolizumab monotherapy during our trial enrollment period. There were 173 patients in the RWD-cetuximab cohort and 658 patients in the RWD-CPI cohort (Table 1). The known prognostic factors including ECOG performance status, number of palliative system therapies, and age that differed significantly across the three cohorts were adjusted using propensity score matching [25]. Based on the propensity score matching, 43 of 45 patients in the combination cohort, 82 of 173 patients in the RWD-cetuximab cohort, and 394 of 658 patients in the RWD-CPI cohort were obtained (Table 4). 

We found no significant difference in the OS between the combination cohort and RWD-CPI cohort (HR 0.91, 95% CI 0.63–1.36, Figure 3A) and between RWD-cetuximab and RWD-CPI cohorts (HR 0.94, 95% CI 0.70–1.29, Figure 3B). In addition, we explored whether there was any difference in the survival outcome depending on the sequence of the monotherapy. Again, the comparison cohorts were adjusted using propensity score matching before the survival analyses [25]. We found no significant OS difference between the patients with (N = 39) or without (N = 355) prior RWD-cetuximab treatments in the RWD-CPI cohort (HR 0.75, 95% CI 0.44–1.20, Figure 3C). However, we found a significant OS difference between the patients with (N = 29) or without (N = 53) prior RWD-CPI in the RWD-cetuximab cohort, favoring no prior RWD-CPI and subsequent receiving RWD-cetuximab (HR 1.81, 95% CI 1.02–3.16, Figure 3D). 

## 3. Discussion

With the success of CPI in the clinics, there has been an intense focus on development of novel immunotherapy approaches and combinations for patients with incurable HNSCC. This has been important since the efficacy of CPI is still very limited, and therapeutic options for patients who fail CPIs are dismal. We explored the combination of cetuximab and nivolumab in attempt to leverage both innate and adaptive immune response against R/M HNSCC. 

Although the cetuximab and nivolumab combination was very well tolerated, the improvement in the 1-year OS did not reach the statistical threshold. These results suggested that the combination of cetuximab and nivolumab might not have the significant synergy as expected compared to the CPI monotherapy considering the majority of our patient cohort (69%) had prior exposure to CPI or cetuximab during the course of their treatments. To provide a clinical context of our data, we also evaluated the OS of two contemporary RWD cohorts, RWD-cetuximab and RWD-CPI, which did not show any significant differences in their OS relative to the OS of the patients with the combination therapy. In addition, even though the median OS was longer in the RWD-CPI cohort (12.8 months) than the RWD-cetuximab cohort (10.5 months), we did not see a statistically significant difference in the OS suggesting there might be an overlap in patients who may benefit from cetuximab or CPI. In a study by Jie et al., cetuximab was shown to promote expansion of circulating and intra-tumoral CTLA4^+^ Treg, and only the patients with stable Treg levels after cetuximab monotherapy responded to the treatment [23]. Presence of Treg is also known to be one of the mechanisms of CPI resistance in HNSCC suggesting overlapping resistance mechanism to the combination therapy [26]. There were two phase III trials showing CPIs were superior in efficacy compared to the investigator’s choice standard of care agents (methotrexate, docetaxel, or cetuximab), but numbers of patients treated with cetuximab monotherapy were limited [7,14]. 

In our study, the subset analyses based on the no prior exposure to CPIs before enrolling in the clinical trial suggested more favorable outcomes, but it did not reach the statistical significance. We further followed up on this issue of the treatment sequence due to the immunomodulatory effects of cetuximab [21,22,23]. A single institution retrospective analysis by Park et al. suggested that cetuximab prior to CPI therapy was associated with worse OS compared to no prior cetuximab [27]. In contrast, a subset analysis of the CheckMate-141 data suggested a superior efficacy of nivolumab over the investigator’s choice standard of care agents regardless of prior exposure to cetuximab [28]. Our results were consistent with the CheckMate-141 showing no OS difference in the RWD-CPI cohort with or without prior RWD-cetuximab treatments. In addition, Park et al. suggested prior CPI treatment might enhance the efficacy of cetuximab [27] while our study suggested prior RWD-CPI was associated with worse OS compared to no prior CPI in the RWD-cetuximab cohort. The difference between the two prior studies [27,28] and our study is that the majority of patients in the two prior studies received cetuximab in combination with chemotherapy (most commonly as EXTREME regimen), and only a fraction of patients received cetuximab monotherapy (1/21 (5%) in Park et al. and 40/147 (27.2%) in CheckMate-141). Considering well established effects of chemotherapy in the immune system [29] and interaction between cetuximab and chemotherapy [9], some of these differences might stem from the different study populations. 

Of interest, we initially saw outcomes favoring cetuximab, but the survival curves intersected favoring the CPI monotherapy in a longer follow up, which is consistent with randomized published data from KN040 and CheckMate-141 trials [7,14]. In addition, several patients with prior failure to CPI achieved durable response to the combination therapy in our trial. Based on our data, it appears that an early introduction of a short term cetuximab treatment prior to CPI to assess the immune modulation in the T cell compartment in circulation and tumors deserves further investigation as well as development of predictive biomarkers for upfront patient selection. For example, the 1-year OS rate of the patients who received the RWD-cetuximab within 30 days of ending RWD-CPI was 38% while the 1-year OS rate of the patients who received the RWD-CPI within 30 days of ending RWD-cetuximab was 55%. Although it was not conclusive due to the small sample size in each group in our study, it is an intriguing result. In CheckMate-141, the information regarding the context of cetuximab use such as the timing relative to on-treatment study or combination with radiation was not available [28]. This approach of using intermittent cetuximab to modulate the immune response for subsequent CPI or early patient selection, particularly in patients who failed earlier CPI, may also alleviate cetuximab-related skin toxicities resulting from a long-term use of cetuximab. 

We fully acknowledge the limitations of using a retrospectively collected RWD. Common limitations are a potential bias in patient selection, differential follow-up time, insufficient number of events, imbalance in co-morbidities, missing data, etc. However, through appropriate study design with aligned cohort selection and data harmonization, significant insight can be gained using a contemporary cohort with specified patient accrual periods and data cutoff dates compared to the use of historical controls. Covariates can be further balanced using appropriate statistical methods such as propensity score matching or weighing [25]. For example, in our study, we selected two contemporary cohorts of RWD-cetuximab and RWD-CPI during our trial enrollment period (December 2017–May 2019). The patients were further selected for information availability of the patient characteristics that will allow us to harmonize with our trial patients including age at treatment, gender, race, ECOG performance status at the start of the treatment, p16 IHC status, line of systemic treatment for R/M HNSCC, primary tumor site, smoking status, start and end dates of cetuximab or nivolumab/pembrolizumab treatments, and survival status. For the number of prior therapy determination, the patients who had more than 90 days of gap between any of treatments were excluded with a concern of receiving additional treatments at institutions outside of the Flatiron Health network and potentially impacting the number of therapy line data. In addition, statistical methods, propensity score matching or weighing, were applied to achieve the required balance. Particularly for the development of immune-oncologic agents, preclinical modeling of the target validation and toxicity and response assessments is extremely limited. Promising agents need to be tested through human clinical trials as soon as possible. Having to enroll a large number of patients to control arms can delay the enrollments, and the cost can be prohibitive. Use of well curated, contemporary RWD cohorts may therefore have a role in early drug development and interpretation of single arm clinical trials.

## 4. Materials and Methods

### 4.1. Study Design and Patient Selection

The single arm, phase I/II study was conducted at three sites including Moffitt Cancer Center, the Ohio State University, and Emory University (NCT03370276). Patients were eligible for enrollment if they met the following criteria: histologically or cytologically confirmed HNSCC of oral cavity, oropharynx, paranasal sinuses, nasal cavity, hypopharynx, or larynx; p16-positive SCC of unknown primary in cervical lymph node; incurable by local therapy such as surgery or radiation therapy with or without chemotherapy; had persistent disease following radiotherapy administered with a chemotherapy sensitizer; progressed on at least one prior line of palliative treatment including chemotherapy, targeted therapy, palliative radiation, immunotherapy, and/or biological therapy regimen for their R/M HNSCC; and had intolerance to standard first-line systemic chemotherapy. Patients were excluded if they met the following criteria: prior exposure to the combination of cetuximab and nivolumab. The detailed eligibility criteria are provided in Appendix A. The trial protocol and all amendments were approved by Institutional Review Board (IRB) at each institution.

The primary objective of the phase I portion of the trial was to determine the safety and tolerability of concurrent cetuximab and nivolumab in patients with R/M HNSCC. The primary objective of the phase II portion of the study was to determine the 1-year OS rate of concurrent cetuximab and nivolumab in patients who had progressed on at least one prior line of palliative treatment for incurable R/M HNSCC. 

During phase I, two dose levels were planned: Dose Level 1 with lead-in cetuximab 500 mg/m^2^ alone (Day-14 before Cycle 1 only) followed by nivolumab 240 mg IV + cetuximab 500 mg/m^2^ every 2 weeks for 24 cycles or discontinuation or Dose Level-1 with lead-in cetuximab 500 mg/m^2^ alone (Day-14 before Cycle 1 only) followed by nivolumab 240 mg IV + cetuximab 250 mg/m^2^ every 2 weeks for 24 cycles or discontinuation (Appendix A). Patients with cetuximab infusion reaction or who did not receive C1D1 for any reason were deemed non-evaluable and replaced. The toxicities with possible, probable, and definite attribution were included in treatment-related adverse events (TRAEs) and immune-related adverse events (IRAEs) analyses. The nivolumab dose reduction was not allowed but withheld/discontinued based on AE severity. Cetuximab dose reduction was allowed and withheld/discontinued based on AE severity. In cases where one agent was withheld both were delayed/resumed concurrently.

### 4.2. Response Assessment

Radiology assessments were obtained every 6 weeks for Cycle 1–4, then every 2 cycles during Cycle 5–6, and then every 3 cycles during Cycle 7–24 while on study drugs. The same types of scans were used for repeat measurements. Response was assessed by RECIST 1.1.

### 4.3. Biomarker Evaluation

The p16 staining was performed using p16 mouse monoclonal antibody predilute, CINtec^®^, clone E6H4 (Roche Tissue Diagnostics, Tucson, AZ, USA) as previously described and scored following the guidelines for p16 interpretation endorsed by the College of American Pathology [4,30,31]. The PD-L1 staining was performed using 22C3 pharmDx assay (Agilent Techologies, Carpinteria, CA, USA) and combined positive score (CPS) was determined as previously described [15]. These biomarkers were evaluated using archived formalin fixed paraffin embedded tumors.

### 4.4. Statistical Assumptions of the Prospective Clinical Trial

To test if the 1-year OS rate of concurrent cetuximab and nivolumab in patients who had progressed on at least one prior line of treatment for their R/M HNSCC exceeded the historical data, we estimated the historical data of nivolumab monotherapy to be 1-year OS 36% [14]. With our expected 56% 1-year OS rate after the combined treatment with both cetuximab and nivolumab, we enrolled 45 patients to reach 90% power to see a statistically significant difference (*p* ≤ 0.05) in OS. In order to be considered evaluable, patients must complete the lead-in period and receive the C1D1 doses of cetuximab and nivolumab. Non-evaluable subjects were replaced.

### 4.5. Patient Characteristics of the Real-World Dataset

The nationwide electronic health record derived RWD from de-identified patients with R/M HNSCC was obtained from Flatiron Health after an approval by the Advarra IRB for the Moffitt Cancer Center, approval date: 28 January 2020—Via Expedite Review, Pro00041669. The Flatiron Health database is a longitudinal database, comprising de-identified patient-level structured and unstructured data, curated via technology-enabled abstraction [32,33]. To provide a clinical context for the degree of the cetuximab and nivolumab combination efficacy, we selected two contemporary cohorts of patients who received their first dose of either cetuximab monotherapy or nivolumab/pembrolizumab monotherapy during our trial enrollment period (December 2017–May 2019). The patients were further selected for availability of the following information: age at treatment, gender, race, ECOG performance status at the start of the treatment, p16 IHC status, line of systemic treatment for R/M HNSCC, primary tumor site, smoking status, start and end dates of cetuximab or nivolumab/pembrolizumab treatments, and survival status. The patients who had more than 90 days of gap between any of treatments were excluded with a concern of receiving additional treatments at institutions outside of the Flatiron Health network and potentially impacting the number of therapy line data.

### 4.6. Statistical Analyses of Prospective Clinical Trial Data and Real-World Data

The clinical features of interests were summarized using descriptive statistics including median and interquartile range for continuous variables and proportions and frequencies for categorical variables. In general, Kruskal–Wallis tests for continuous variables and chi-squared tests for categorical variables were conducted to compare the differences among the multiple groups. For those categorial variables in which some levels were less than 5 counts, the Fisher’s exact test was applied.

For survival analyses of the prospective clinical trial, progression-free survival (PFS) time was defined as the time between the date of study enrollment and the date of progressive disease or death whichever happened first or otherwise censored at the last date of known alive. Overall survival (OS) time was defined as the time between the date of study enrollment and the date of death or censored at the last date known alive. For survival analyses of the RWD, PFS time was not available. The OS was defined as the time between the date of first dose of given treatments and the date of death or censored at the last date known alive. The propensity score matching [25] was applied to match our prospective clinical trial and the RWD based on the prognostic factors which are significantly different across the three cohorts: cetuximab and nivolumab combination, cetuximab monotherapy (RWD-cetuximab), and RWD-nivolumab/pembrolizumab CPI monotherapy (RWD-CPI). For subset analyses of the RWD, propensity score weighing [25] was used for given 2-group comparisons. The Kaplan–Meier method was used for the PFS and OS analyses, and log-rank tests were adopted to compare survival differences between two groups. Univariate Cox proportional hazards (PH) model was conducted to evaluate the association of OS with the sequential treatment or the individual clinical feature. All statistical analyses were performed using SAS (version 9.4, SAS Institute Inc., Cary, NC, USA) and the R 3.6.0 software (https://www.R-project.org, accessed on 24 November 2020). 

## 5. Conclusions

Only a limited number of patients with R/M HNSCC benefit from CPI. In this study, we evaluated toxicities and efficacy of a combination of cetuximab and nivolumab and potential effects of sequential administration of CPI and cetuximab. We also evaluated electronic health record-derived RWD from patients who were treated RWD-cetuximab monotherapy and RWD-CPI monotherapy to provide clinical context for our prospective findings and to explore sequential treatment effects of cetuximab and CPI. This is, to our knowledge, the first report describing the combination of cetuximab and nivolumab in R/M HNSCC. Our data suggest the combination of cetuximab and nivolumab has a tolerable toxicity profile. Although patients with no prior CPI showed a trend for more favorable PFS relative to patients with prior CPI, and this regimen was active in some patients with prior CPI failure, the OS did not meet our statistical assumption in this heavily pre-treated population. Based on our analyses of contemporary RWD cohorts, the patients treated with the RWD-cetuximab after RWD-CPI failure have a poor prognosis. Despite the fact that our study was not powered to answer a sequencing question, the results open the door for further evaluation of sequential approaches in a selected subset of patients.

## Figures and Tables

**Figure 1 cancers-13-01180-f001:**
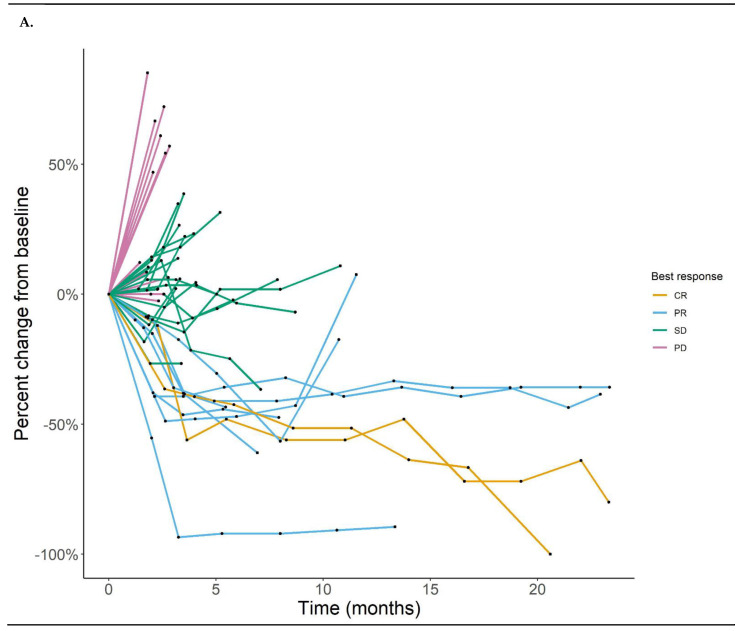
(**A**) Spider plot of best response. (**B**) Waterfall plot of best response. “+” indicates prior exposure to anti-PD-1 checkpoint inhibitors (CPI). “-” indicates no prior exposure to anti-PD-1 CPI.

**Figure 2 cancers-13-01180-f002:**
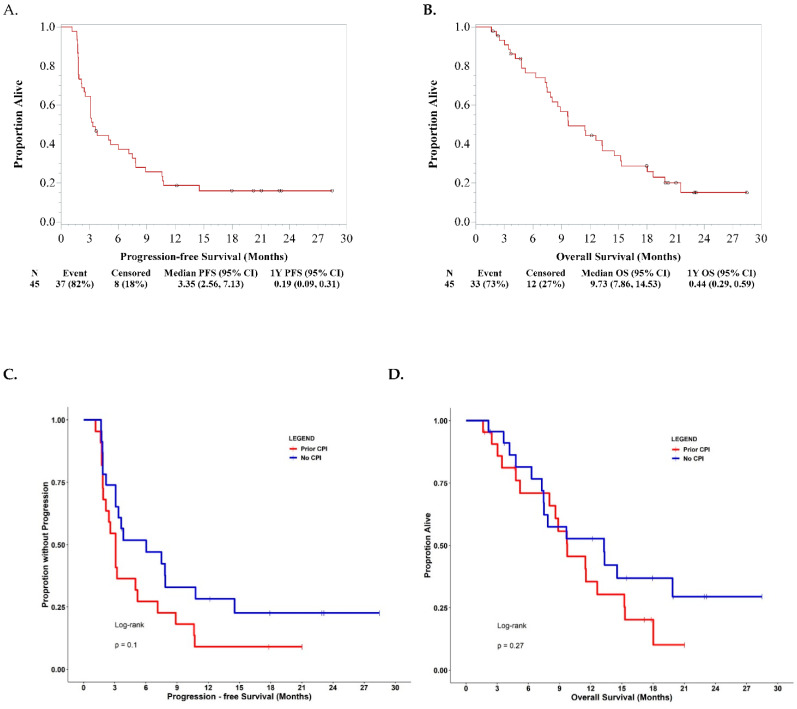
(**A**) Progression-free survival (PFS) given the combination of cetuximab and nivolumab. (**B**) Overall survival (OS) given the combination of cetuximab and nivolumab. (**C**) PFS based on exposure history to anti-PD-1 checkpoint inhibitors (CPI). (**D**) OS based on exposure history to anti-PD-1 CPI.

**Figure 3 cancers-13-01180-f003:**
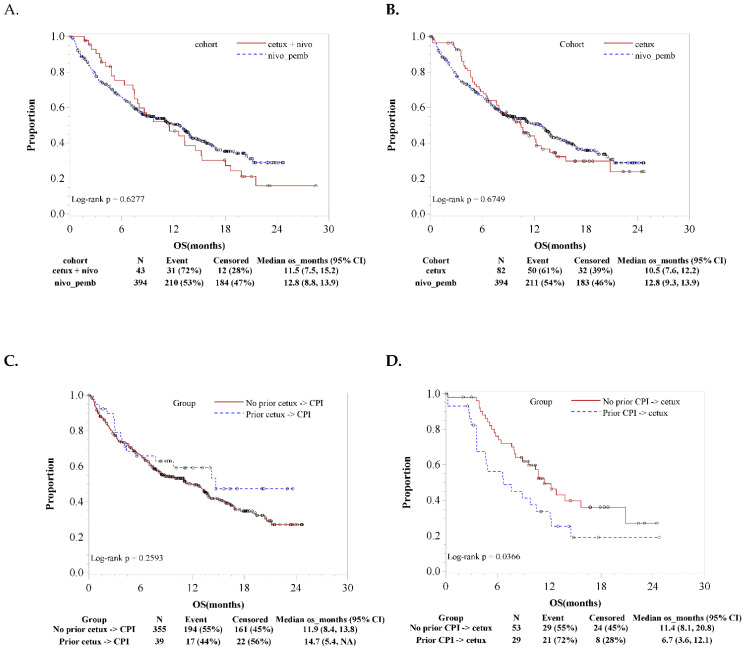
(**A**) Overall survival (OS) comparison between the combination of cetuximab and nivolumab in the clinical trial cohort and nivolumab or pembrolizumab checkpoint inhibitor (CPI) monotherapy in the real-world data (RWD-CPI) cohort. (**B**) OS comparison between the RWD-cetuximab cohort and RWD-CPI cohort. (**C**) OS comparison between the patients who received RWD-cetuximab and the patients who did not receive RWD-cetuximab before receiving RWD-CPI. (**D**) OS comparison between the patients who received RWD-CPI and the patients who did not receive RWD-CPI before receiving RWD-cetuximab. All analyses were conducted after adjusting the clinical characteristics of the cohorts using propensity score matching.

**Table 1 cancers-13-01180-t001:** Patient Characteristics.

	Cetux + Nivo N = 45	Rwd-Cetuximab N = 173	Rwd-Cpi N = 658	Overall *p*-Value
Age_at_treatment	64.0 [57.0;68.0]	68.0 [62.0;75.0]	66.0 [59.0;72.0]	0.001
Gender				0.098
F	8 (17.8%)	48 (27.7%)	135 (20.5%)	
M	37 (82.2%)	125 (72.3%)	523 (79.5%)	
Race				<0.001
White	40 (88.9%)	93 (53.8%)	445 (67.6%)	
Black or African American	3 (6.67%)	16 (9.2%)	42 (6.4%)	
Others	2 (4.44%)	64 (37.0%)	171 (26.0%)	
Ecog PS				0.097
0	9 (20.0%)	45 (26.0%)	182 (27.7%)	
1	33 (73.3%)	93 (53.8%)	351 (53.3%)	
2	3 (6.7%)	35 (20.2%)	125 (19.0%)	
p16 Status				0.762
non-OP (pos + neg + unknown)	19 (42.2%)	89 (51.4%)	309 (47.0%)	
OP (neg + unknown)	4 (8.9%)	10 (5.8%)	47 (7.1%)	
OP and unknown primary (pos)	22 (48.9%)	74 (42.8%)	302 (45.9%)	
Line of systemic treatment				<0.001
1	11 (24.4%)	92 (53.2%)	257 (39.1%)	
2	27 (60.0%)	43 (24.9%)	302 (45.9%)	
3	5 (11.1%)	25 (14.5%)	86 (13.1%)	
4	2 (4.4%)	13 (7.5%)	13 (2.0%)	
Primary tumor site				0.703
Hypopharynx	3 (6.7%)	12 (6.9%)	45 (6.8%)	
Larynx	6 (13.3%)	46 (26.6%)	138 (21.0%)	
Oral cavity	10 (22.2%)	31 (17.9%)	126 (19.1%)	
Oropharynx	24 (53.3%)	79 (45.7%)	333 (50.6%)	
Unknown Primary	2 (4.4%)	5 (2.9%)	16 (2.4%)	
Smoking status				0.319
History of smoking	33 (73.3%)	144 (83.2%)	531 (80.7%)	
No history of smoking	12 (26.7%)	29 (16.8%)	127 (19.3%)	
Survival status				0.066
alive	12 (26.7%)	71 (41.0%)	290 (44.1%)	
death	33 (73.3%)	102 (59.0%)	368 (55.9%)	
Length of therapy(weeks)	14.0 [8.00; 32.0]	19.9 [11.0; 32.1]	17.9 [8.00;40.8]	0.879

CETUX: cetuximab, NIVO: nivolumab, RWD-Cetuximab: Real World Data-cetuximab monotherapy, RWD-CPI: Real World Data-nivolumab or pembrolizumab checkpoint inhibitor monotherapy, ECOG PS: Eastern Cooperative Oncology Group Performance Status, OP: oropharynx.

**Table 2 cancers-13-01180-t002:** Treatment-Related Adverse Events.

Toxicity	Toxicity Category	Grade 3 N (%)	Grade 4 N (%)
Alkaline phosphatase increased	Investigations	1 (2.2)	-
Anemia	Blood and lymphatic system disorders	1 (2.2)	-
Dizziness	Nervous system disorders	1 (2.2)	-
Dyspnea	Respiratory, thoracic and mediastinal disorders	1 (2.2)	-
Fatigue	General disorders and administration site conditions	6 (13.3)	-
General disorders and administration site conditions—Other, specify	General disorders and administration site conditions	-	1 (2.2)
Generalized muscle weakness	Musculoskeletal and connective tissue disorders	1 (2.2)	-
Hypertension	Vascular disorders	1 (2.2)	-
Hypomagnesemia	Metabolism and nutrition disorders	1 (2.2)	-
Hyponatremia	Metabolism and nutrition disorders	1 (2.2)	-
Myocarditis	Cardiac disorders	1 (2.2)	-
Palmar-plantar erythrodysesthesia syndrome	Skin and subcutaneous tissue disorders	1 (2.2)	-
Rash acneiform	Skin and subcutaneous tissue disorders	2 (4.4)	-
Rash maculo-papular	Skin and subcutaneous tissue disorders	1 (2.2)	-
White blood cell decreased	Investigations	1 (2.2)	-
Overall		13 (28.9)	1 (2.2)

**Table 3 cancers-13-01180-t003:** Immune-Related Adverse Events.

Toxicity	Toxicity Category	Grade 3 N (%)	Grade 4 N (%)
Alkaline phosphatase increased	Investigations	1 (2.2)	-
Anemia	Blood and lymphatic system disorders	1 (2.2)	-
Aspartate aminotransferase increased	Investigations	1 (2.2)	-
Dizziness	Nervous system disorders	1 (2.2)	-
Dyspnea	Respiratory, thoracic and mediastinal disorders	1 (2.2)	-
Hyponatremia	Metabolism and nutrition disorders	1 (2.2)	-
Myocardial infarction	Cardiac disorders	1 (2.2)	-
Myocarditis	Cardiac disorders	1 (2.2)	-
Rash maculo-papular	Skin and subcutaneous tissue disorders	1 (2.2)	-
Fatigue	General disorders and administration site conditions	3 (6.7)	-
Overall		7 (15.6)	0 (0.0)

**Table 4 cancers-13-01180-t004:** Patient characteristics after propensity score matching.

	Cetux + Nivo N = 43	Rwd-Cetuximab N = 82	Rwd-Cpi N = 394	Overall *p*-Value
**Age_at_treatment**	64.0 [57.0;68.0]	65.0 [59.0;70.0]	64.5 [58.0;70.0]	0.692
**Gender**				0.106
F	7 (16.3%)	24 (29.3%)	77 (19.5%)	
M	36 (83.7%)	58 (70.7%)	317 (80.5%)	
**Race**				0.002
White	38 (88.4%)	44 (53.7%)	272 (69.0%)	
Black or African American	3 (7.0%)	7 (8.5%)	20 (5.1%)	
Others	2 (4.7%)	31 (37.8%)	102 (25.9%)	
**Ecog PS**	1.00 [1.00;1.00]	1.00 [0.00;1.00]	1.00 [1.00;1.00]	0.504
**p16 Status**				0.685
non-OP (pos + neg + unknown)	17 (39.5%)	41 (50.0%)	187 (47.5%)	
OP (neg + unknown)	4 (9.3%)	3 (3.7%)	25 (6.4%)	
OP and unknown primary (pos)	22 (51.2%)	38 (46.3%)	182 (46.2%)	
**Line of systemic treatment**	2.00 [1.50;2.00]	2.00 [1.00;2.00]	2.00 [1.00;2.00]	0.515
**Primary tumor site**				0.851
Hypopharynx	3 (6.98%)	6 (7.32%)	34 (8.63%)	
Larynx	6 (14.0%)	22 (26.8%)	79 (20.1%)	
Oral cavity	8 (18.6%)	13 (15.9%)	74 (18.8%)	
Oropharynx	24 (55.8%)	38 (46.3%)	196 (49.7%)	
Unknown Primary	2 (4.65%)	3 (3.66%)	11 (2.79%)	
**Smoking status**				0.751
History of smoking	33 (76.7%)	66 (80.5%)	321 (81.5%)	
No history of smoking	10 (23.3%)	16 (19.5%)	73 (18.5%)	
**Survival status**				0.039
alive	12 (27.9%)	32 (39.0%)	184 (46.7%)	
death	31 (72.1%)	50 (61.0%)	210 (53.3%)	
**Length of therapy** **(weeks)**	14.0 [8.00;32.9]	17.7 [9.14;31.5]	17.9 [9.00;40.1]	0.713

CETUX: cetuximab, NIVO: nivolumab, RWD-Cetuximab: Real World Data-cetuximab monotherapy, RWD-CPI: Real World Data-nivolumab or pembrolizumab checkpoint inhibitor monotherapy, ECOG PS: Eastern Cooperative Oncology Group Performance Status, OP: oropharynx.

## Data Availability

The data presented in this study are available within the manuscript and the Appendix A.

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
