# Peer review of "Concurrent Cetuximab and Nivolumab as a Second-Line or beyond Treatment of Patients with Recurrent and/or Metastatic Head and Neck Squamous Cell Carcinoma: Results of Phase I/II Study"

_cancers, 2021, doi:10.3390/cancers13051180_

Round 1

Reviewer 1 Report

Chung et al. presented the results of a phase I/II trial exploring the combination of cetuximab and nivolumab in recurrent/metastatic (R/M) head and neck squamous cell carcinoma (HNSCC). A retrospective analysis of RWD was conducted as well.

The concurrent administration of EGFR-inhibitors and anti-PD-1 agents is an interesting combination that deserves to be explored in R/M HNSCC. However, the present study has some major limitations: 

  1. it is not clear how many patients received platinum-based chemotherapy and in which context (concomitantly with radiation? as first-line palliative treatment? both?). This aspect is fundamental in HNSCC, especially in R/M disease, and cannot be omitted in the presentation of the results and in the discussion
  2. the study was performed as second-line treatment (or further lines). This should be clearly stated in the title and in the background. 
  3. the 3 patient populations that were analyzed to discuss the results are too different. In particular, in the cetux+nivo arm 24.4% of patients received only one line of previous treatment, this means that we are dealing with a quite heavily pre-treated population. On the other side, the frequency of patients having received at least 2 lines of treatment was 46.8% in RWD-cetuximab and 60.9% in RWD-CPI. This imbalance should not be neglected while interpreting the results and should be discussed more extensively.

Overall the methodology is well discussed and the study has interesting and hypothesis-generating aspects. However, the aforementioned limitations of the study should be discussed more extensively and critically. 

After major revision, this manuscript might be more suitable for journal more focused on head and neck cancers. 

Reviewer 2 Report

This is a well written manuscript on a series of patients treated with sequential cetuximab and  nivolumab.

The authors want to compare their outcome tore world dataset of patients treated with mono therapy. Although this becomes clear in reading, the abstract is not clear about which datasets or comparisons were made. Please clarify in the abstract.

It remains questionable whether it is scientifically sound to compare different trials with real-life data. The authors should ad some comments on possible biases using this methodology.

The manuscript is well documented and tables are clear. The series is quite small.

Reviewer 3 Report

Chung CH et al conducted a phase I/II study to evaluate the safety and efficacy of the cetuximab-nivolumab combination in incurable recurrent and/or metastatic HNSCC. They also used electronic health record-derived real world data to explore sequential treatment effects of cetuximab and ICI. A randomized phase II study comparing concurrent with sequential administration of these 2 agents would be preferable to address this question. However, it would be difficult to recruit pts given the approval of chemo+pembro in 1st line for all comers in the US. The authors reach the correct conclusion that sequencing may matter but their study can not provide an answer.

Major comments:

  1. In the eligibility criteria of phase I/II portion prior cetuximab or ICI exposure was allowed. As a result, the efficacy of the combo can not be evaluated with confidence and it is not surprising that the improvement in 1-year OS did not reach the stat threshold.
  2. The use of the retrospectively collected RWD cohort as a comparator has several limitations including comorbidities and other factors that have an impact on OS.

Round 2

Reviewer 1 Report

The authors addressed all the points commented in previous revision.